# The PINK1 Activator Niclosamide Mitigates Mitochondrial Dysfunction and Thermal Hypersensitivity in a Paclitaxel-Induced *Drosophila* Model of Peripheral Neuropathy

**DOI:** 10.3390/biomedicines10040863

**Published:** 2022-04-07

**Authors:** Hye-Ji Jang, Young-Yeon Kim, Kang-Min Lee, Jung-Eun Shin, Jeanho Yun

**Affiliations:** 1Peripheral Neuropathy Research Center, Dong-A University, Busan 49201, Korea; rainbow5449@naver.com (H.-J.J.); kmlee8348@dau.ac.kr (K.-M.L.); jeshin20@dau.ac.kr (J.-E.S.); 2Department of Biochemistry, College of Medicine, Dong-A University, Busan 49201, Korea; 3Department of Translational Biomedical Sciences, Graduate School of Dong-A University, Busan 49201, Korea; 4Department of Molecular Neuroscience, College of Medicine, Dong-A University, Busan 49201, Korea

**Keywords:** peripheral neuropathy, paclitaxel, niclosamide, PINK1, mitochondrial dysfunction

## Abstract

Paclitaxel is a widely used anticancer drug that induces dose-limiting peripheral neuropathy. Mitochondrial dysfunction has been implicated in paclitaxel-induced neuronal damage and in the onset of peripheral neuropathy. We have previously shown that the expression of PINK1, a key mediator of mitochondrial quality control, ameliorated the paclitaxel-induced thermal hyperalgesia phenotype and restored mitochondrial homeostasis in *Drosophila* larvae. In this study, we show that the small-molecule PINK1 activator niclosamide exhibits therapeutic potential for paclitaxel-induced peripheral neuropathy. Specifically, niclosamide cotreatment significantly ameliorated the paclitaxel-induced thermal hyperalgesia phenotype in *Drosophila* larvae in a PINK1-dependent manner. Paclitaxel-induced alteration of the dendrite structure of class IV dendritic arborization (C4da) neurons was not reduced upon niclosamide treatment. In contrast, paclitaxel treatment-induced increases in both mitochondrial ROS and aberrant mitophagy levels in C4da neurons were significantly suppressed by niclosamide. In addition, niclosamide suppressed paclitaxel-induced mitochondrial dysfunction in human SH-SY5Y cells in a PINK1-dependent manner. These results suggest that niclosamide alleviates thermal hyperalgesia by attenuating paclitaxel-induced mitochondrial dysfunction. Taken together, our results suggest that niclosamide is a potential candidate for the treatment of paclitaxel-induced peripheral neuropathy with low toxicity in neurons and that targeting mitochondrial dysfunction is a promising strategy for the treatment of chemotherapy-induced peripheral neuropathy.

## 1. Introduction

Paclitaxel is a widely used anticancer drug for many human solid tumors including breast, ovarian, and lung cancers [1,2]. However, nerve damage leading to painful peripheral neuropathy occurs at high frequency and limits the clinical application of paclitaxel [3]. The major side effects are predominantly associated with sensory nerve abnormalities such as allodynia and hyperalgesia [4]. These symptoms often persist for several months to years and some side effects affect the lifetime of cancer survivors [5]. Although paclitaxel-induced peripheral neuropathy has profound impacts on the duration of chemotherapy and the quality of life of cancer patients, there is no effective treatment to prevent or reverse damage to sensory neurons.

Mitochondrial dysfunction has been implicated in chemotherapy-induced peripheral neuropathy (CIPN) caused by various anticancer drugs, including paclitaxel [4,6,7,8,9]. Previous studies revealed that paclitaxel induces various signs of mitochondrial dysfunction, such as mitochondrial membrane potential loss, mitochondrial reactive oxygen species (ROS) increase, ATP production reduction, and mitochondrial swelling, in both in vitro and in vivo models [6,8,10,11,12]. Given that mitochondria are fundamental to many important functions of neurons, including energy generation, survival, neuronal plasticity, and resistance to stresses [13,14], it is not surprising that mitochondrial dysfunction is considered a major contributor to the onset and progression of CIPN [4,15]. Therefore, the development of a strategy to improve or reduce mitochondrial dysfunction is important for CIPN treatment.

We had previously shown that sensory neuron-specific expression of PINK1 significantly ameliorated the thermal hyperalgesia phenotype in a paclitaxel-induced *Drosophila* larvae CIPN model [16]. Interestingly, PINK1 expression did not suppress paclitaxel-induced alterations to the sensory dendrite structure of class IV dendritic arborization (C4da) neurons which are responsible for behavioral responses to various noxious stimuli, including high temperatures [17,18]. However, paclitaxel-induced increases in both mitochondrial ROS and mitophagy were significantly suppressed by PINK1 expression [16], suggesting that PINK1 can reduce the mitochondrial dysfunction induced by paclitaxel. Although we observed that PINK1 expression ameliorated paclitaxel-induced mitochondrial dysfunction and attenuated the thermal hyperalgesia phenotype, our previous results also indicated that constitutive PINK1 overexpression led to several side effects. For example, PINK1 expression induced alterations to the dendrite structure of C4da neurons. In addition, thermal nociception interfered with PINK1 expression. Thus, transient pharmacological activation of PINK1 is a preferable treatment strategy.

In this study, we examined the effect of niclosamide, which has been recently reported to be a PINK1 activator [19]. In a *Drosophila* larval model, niclosamide ameliorated the paclitaxel-induced thermal hyperalgesia phenotype in a PINK1-dependent manner. Moreover, niclosamide improved mitochondrial dysfunction induced by paclitaxel treatment in both *Drosophila* and human neuronal cell lines. Our results confirm that PINK1 is a potential target for CIPN treatment and suggest that niclosamide is a candidate for the treatment of paclitaxel-induced peripheral neuropathy.

## 2. Materials and Methods

### 2.1. Drosophila Strains and Treatments

The *w^1118^*, *ppk-GAL4*, *ppk^1a^-GAL4*, *UAS-CD4-tdTomato* (*CD4-tdTom*), *UAS-mt-Keima*, *PINK1* RNAi, *UAS-GFP dsRNA* (BL9331), and *UAS-mito-roGFP2-Orp1* lines were described previously [16].

Paclitaxel (Cayman Chemical Co., Ann Arbor, MI, USA) was administered following a feeding regimen described previously [16]. Briefly, twenty virgin female flies were mated with fifty male flies for 48–72 h, and the embryos were collected for 2–4 h on grape juice agar plates. The embryos were grown for 72 h to develop into L3 larvae. The larvae were rinsed with distilled water and transferred to freshly prepared grape juice agar plates containing either 20 μM paclitaxel or 0.2% EtOH. The larvae were grown for another 48 h before performing the thermal nociception assay. Niclosamide (Calbiochem, San Diego, CA, USA) (250 μM) was administered together with paclitaxel using EtOH as the vehicle.

### 2.2. Thermal Nociception Assays

Thermal nociception assays were performed as described previously [16]. Briefly, L3 larvae (120 h after egg laying [AEL]) were rinsed with distilled water and gently placed on a Petri dish. After 10 sec of acclimation, the larval abdominal A4–A5 segments were touched under a microscope with a custom-built 0.6 mm-wide thermal probe whose temperature was controlled by a microprocessor. The time required to induce the aversive corkscrew-like rolling response was measured as the withdrawal latency with a 20 s cut-off. The larvae showing no rolling response within 20 s were considered to have no response. For each thermal nociception assay, at least 30 larvae were analyzed. The thermal nociception assay was repeated three times, and the results are presented as the mean values with the standard errors of the means (S.E.M.).

The larval size was measured as described previously [16] after the thermal nociception assay. The larval area was calculated using ImageJ software (NIH, Bethesda, MD, USA), and at least 30 larvae per sample group were measured.

### 2.3. Analysis of the Images of C4da Neuron Dendrites

The analysis of the dendritic structure of the C4da neurons of the L3 larvae was performed as described previously [16] using a Zeiss LSM 800 confocal microscope (Carl Zeiss, Oberkochen, Germany) at the Neuroscience Translational Research Solution Center. The dorsal projection of one C4da neuron per larva from abdominal segment A4 was analyzed. At least 3–4 larvae in each genotype group were analyzed. The dendrite image analysis was repeated three times, and the results are presented as the mean values with the S.E.M.

### 2.4. In Vivo Measurement of Mitochondrial ROS in C4da Neurons

For in vivo ROS imaging, *ppk>mito-roGFP2-Orp1* L3 larvae were examined with a Zeiss LSM 800 confocal microscope (Carl Zeiss) at the Neuroscience Translational Research Solution Center with a 405-nm (oxidized) or 488-nm (reduced) excitation laser using 520-nm emission as previously described [20]. One C4da neuron per larva from abdominal segment A4 was analyzed. The 405-nm/488-nm fluorescence intensity was obtained using Zeiss Zen software, and five larvae in each group were measured. The experiment was repeated three times, and the results are presented as the mean values with S.E.M.

### 2.5. Measurement of Mitophagy Levels

Mitophagy levels of C4da neurons at abdominal segment A4 of L3 larvae were measured as described previously [16] using a Zeiss LSM 800 confocal microscope at the Neuroscience Translational Research Solution Center. The mitophagy level was quantified through a pixel-by-pixel analysis of mt-Keima confocal microscopy images using Zeiss Zen software as described previously [16,21]. One C4da neuron per larva from abdominal segment A4 was analyzed. To quantify the mitophagy level in the C4da neurons, five larval samples were used for quantification, and the average values were calculated. The results are presented as the mean values with the SD.

### 2.6. Cell Lines, RNA Interference and Treatments

SH-SY5Y cells were maintained in DMEM containing 10% fetal bovine serum (FBS; JR Scientific Inc., Woodland, CA, USA). To knockdown PINK1 expression, SH-SY5Y cells were infected with either a lentivirus encoding PINK1 shRNA (Sigma-Aldrich Inc., St. Louis, MO, USA, TRCN0000199193) or a control lentivirus and selected with 2 μg/mL puromycin. SH-SY5Y cells were treated with paclitaxel (1 μM) for 24 h, and then treat with either niclosamide (10 μM) or EtOH for 9 h. The cells were cultured for another 15 h before performing mitochondrial ROS analysis.

### 2.7. Western Blot Analysis and Antibodies

Cells were lysed in RIPA buffer and subjected to Western blot analysis as described previously [22]. An anti-OpaI antibody was obtained from BD Biosciences (San Jose, CA, USA). Anti-Parkin pS65, anti-Parkin, anti-actin antibodies were purchased from Cell Signaling Technology Inc. (Danvers, MA, USA). An anti-PINK1 antibody was obtained from Novus Biologicals (Centennial, CO, USA).

### 2.8. Measurement of Mitochondrial ROS Levels in the Cell Lines

To measure the mitochondrial ROS levels, cells were harvested and then stained with the mitochondrial superoxide indicator MitoSOX^TM^ Red (5 μM) (Invitrogen, Carlsbad, CA, USA) for 30 min. The fluorescence intensities were quantified via flow cytometry (Attune NxT Flow Cytometer, Thermo Fisher Scientific, Waltham, MA, USA) at the Neuroscience Translational Research Solution Center. The results obtained from three repeated experiments are presented as the mean with SD.

### 2.9. Statistical Analysis

The results are presented as the mean ± SD or mean ± S.E.M. as indicated in the figure legend. One-way ANOVA with Sidák correction was used to compare three or more groups. All data were statistically analyzed using GraphPad Prism software (version 8.0: GraphPad Software, Inc., La Jolla, CA, USA). A *p* value of <0.05 was considered statistically significant.

### 2.10. Genotypes

The following genotypes were used: *ppk>w^1118^* (*ppk-GAL4/+*); *ppk^1a^>CD4-tdTom* (*ppk^1a^-GAL4/UAS-CD4-tdTomato*); *ppk>GFP RNAi* (*ppk-GAL4/UAS-GFP RNAi*; *tub-GAL80ts*), *ppk>PINK1 RNAi* (*ppk-GAL4/UAS-PINK1 RNAi*; *tub-GAL80ts*); *ppk^1a^>mt-Keima* (*ppk^1a^-GAL4*, *UAS-mt-Keima/+*); *ppk>mito-roGFP2-Orp1* (*ppk-GAL4/UAS-mito-roGFP2-Orp1*).

## 3. Results

### 3.1. Niclosamide Ameliorates the Paclitaxel-Induced Heat Hyperalgesia Phenotype in Drosophila Larvae

To test the therapeutic potential of niclosamide on paclitaxel-induced peripheral neuropathy, we examined the effect of niclosamide on the thermal nociceptive response using a previously established paclitaxel-induced thermal hypersensitivity model in *Drosophila* larvae [16,23]. Consistent with previous reports [16,23], paclitaxel treatment (20 μM) in our study resulted in a significant decrease in the time required to induce the aversive corkscrew-like rolling response (mean withdrawal latency; MWL) in the larval thermal nociception assay. Upon paclitaxel treatment, the MWL from the 40 °C heat probe decreased from 7.5 s to 3.8 s, indicating that a heat hyperalgesia phenotype had developed upon paclitaxel treatment (Figure 1A). Importantly, the thermal nociception of *Drosophila* larvae cotreated with niclosamide (250 μM) was not changed upon paclitaxel treatment, as evidenced by the absence of a significant change in the MWL, and niclosamide treatment alone did not significantly reduce the MWL. These results suggest that niclosamide significantly alleviated the paclitaxel-induced thermal hypersensitivity phenotype.

Niclosamide has been used for a long time as an anthelminthic drug because it induces no apparent side effects in humans [24]. Confirming that niclosamide caused no adverse effects on the larvae, the size of larvae treated with niclosamide for 48 h was not different from that of control larvae (Figure 1B), indicating that niclosamide does not significantly interfere with larval growth.

### 3.2. The Effect of Niclosamide on the Paclitaxel-Induced Heat Hyperalgesia Phenotype Is PINK1-Dependent

To verify that niclosamide suppressed the paclitaxel-induced thermal hyperalgesia phenotype through PINK1, we examined the effect of niclosamide on the specific knockdown of PINK1 expression in C4da neurons because C4da neurons are responsible for behavioral responses to high temperatures [17,18]. We had previously observed in a heat probe assay that knocking down PINK1 by using PINK shRNA in C4da neurons resulted in a significant decrease in MWL [16]. To observe the effect of niclosamide on the specific knockdown of PINK1 expression more clearly, we moderately inhibited PINK1 expression by using a temperature-sensitive mutant of the GAL80 repressor GAL4, GAL80ts [25]. GAL80ts is active at 18 °C but inactivated at 29 °C, so it could not repress GAL4-dependent expression. To induce moderate inhibition of PINK1 expression, we maintained larvae expressing PINK1 RNAi (*ppk>PINK1 RNAi*; *tub-GAL80ts*) at 27 °C instead of 29 °C after paclitaxel with niclosamide treatment. Under this moderate PINK1 knockdown condition, the MWL to the 40 °C heat probe was slightly reduced (from 7.0 to 5.9 s). Importantly, although PINK1 expression was moderately inhibited, the effect of niclosamide on the paclitaxel-induced reduction in MWL was completely abolished, and the paclitaxel-induced reduction in MWL was significantly reduced by niclosamide in the control RNAi larvae (*ppk>GFP RNAi*; *tub-GAL80ts*) (Figure 2). These results suggest that the effect of niclosamide on paclitaxel-induced thermal hypersensitivity was dependent on PINK1.

### 3.3. Niclosamide Does Not Reduce Paclitaxel-Induced Alterations in C4da Neuron Arborization

Alterations in the sensory dendritic structure of C4da neurons are associated with paclitaxel-induced peripheral neuropathy [16,23]. We next analyzed the effect of niclosamide on paclitaxel-induced alterations of dendrite structure in C4da neurons. We had previously shown that PINK1 expression did not suppress structural alterations upon paclitaxel treatment [16]. Consistent with this result, analysis of terminal dendrite morphology and branch points in C4da neurons performed by visualizing the C4da neuron plasma membrane marker CD4-tdTomato revealed that paclitaxel-induced alterations in the dendrite length and the number of dendrite branch points were not affected by niclosamide (Figure 3A,B). Paclitaxel treatment alone increased the dendrite length by 26% and the number of dendrite branch points by 42%. Similarly, cotreatment with niclosamide and paclitaxel increased the dendrite length by 28% and the number of dendrite branch points by 31%. Interestingly, whereas PINK1 expression itself significantly reduced dendrite arborization of C4da neurons in our previous study [16], here we observed that niclosamide treatment alone did not significantly change either the dendrite length or the number of dendrite branch points in C4da neurons (Figure 3A,B). In conjunction with the results showing that thermal nociception did not change upon niclosamide treatment, these results indicate that niclosamide is not toxic to C4da sensory neurons.

### 3.4. Niclosamide Alleviates Paclitaxel-Induced Mitochondrial Dysfunction in C4da Neurons

Our previous results had suggested that PINK1 alleviates paclitaxel-induced thermal hypersensitivity by improving mitochondrial dysfunction [16]. To determine the effect of niclosamide on paclitaxel-induced mitochondrial dysfunction, we examined the level of mitochondrial ROS levels in C4da neurons upon paclitaxel and niclosamide cotreatment by expressing the in vivo mitochondrial ROS probe mito-roGFP2-Orp1 [20] specifically in C4da neurons. As shown in Figure 4, the increase in mitochondrial ROS levels in response to paclitaxel treatment was significantly reduced by niclosamide cotreatment.

We next examined the effect of niclosamide on the induction of mitophagy, which is a marker for mitochondrial dysfunction [21,26], upon paclitaxel treatment. The level of mitophagy in C4da neurons was measured quantitatively by expressing the pH-sensitive mitophagy fluorescent probe mt-Keima [26] specifically in C4da neurons. Measurement of mitophagy levels by analyzing the mt-Keima signal revealed that paclitaxel increased the mitophagy level of C4da neurons by approximately 3.7-fold (Figure 5A,B) indicating that paclitaxel induced mitochondrial dysfunction in C4da neurons. Niclosamide cotreatment significantly reduced the increase in mitophagy levels in response to paclitaxel treatment. Notably, the mitophagy level in C4da neurons was not significantly changed upon niclosamide treatment alone (Figure 5B), suggesting that niclosamide did not induce mitochondrial stress, at least under this experimental condition. Taken together, these results suggest that niclosamide reduced paclitaxel-induced mitochondrial dysfunction in C4da neurons.

### 3.5. Niclosamide Ameliorates Paclitaxel-Induced Mitochondrial Dysfunction in SH-SY5Y Cells in a PINK1-Dependent Manner

To further validate the effect of niclosamide on paclitaxel-induced mitochondrial dysfunction, we tested the effect of niclosamide in the human neuroblastoma cell line SH-SY5Y. It has been recently shown that niclosamide activates PINK1 through a temporal and reversible decrease in the mitochondrial membrane potential [19]. Notably, niclosamide treatment (10 μM) for 9 h resulted in an increase in both Parkin Ser65 phosphorylation, a marker of PINK1 activation [27], and cleavage of OPA1, a marker of mitochondrial membrane uncoupling [28] (Figure 6A). Parkin Ser65 phosphorylation upon niclosamide treatment was abolished by PINK1 knockdown, while OPA1 cleavage was not changed, confirming that niclosamide-mediated activation of PINK1 can be suppressed by expressing PINK1 shRNA (Figure 6A).

We next examined the effect of niclosamide on paclitaxel-induced mitochondrial dysfunction in SH-SY5Y cells. As shown in Figure 6B, niclosamide cotreatment completely suppressed the increase in mitochondrial ROS levels in response to paclitaxel treatment. However, the expression of PINK1 shRNA abolished the effect of niclosamide on the paclitaxel-induced increase in mitochondrial ROS levels (Figure 6B). These results suggest that niclosamide efficiently suppressed the paclitaxel-induced increase in mitochondrial ROS in a PINK1-dependent manner in human cells.

## 4. Discussion

In this study, we report that the PINK1 activator niclosamide exhibits therapeutic potential in paclitaxel-induced peripheral neuropathy. We show that cotreatment with niclosamide ameliorated the paclitaxel-induced thermal hyperalgesia phenotype in a Drosophila larval model. While paclitaxel treatment reduced the time required to respond to the heat probe (mean withdrawal latency; MWL) by nearly half, niclosamide completely suppressed paclitaxel-induced sensitivity to heat sensing in Drosophila larvae. We confirmed that the effect of niclosamide on the paclitaxel-induced thermal hyperalgesia phenotype was abolished by PINK1 expression knockdown in C4da neurons, suggesting that niclosamide exerts its neuroprotective function through PINK1.

Niclosamide is a well-known anthelminthic drug used to treat parasite infections in several million people worldwide [24]. Since receiving US FDA approval for use in humans to treat tapeworm infection in 1982, niclosamide has been widely used without causing severe side effects [29]. Interestingly, recent studies have shown that niclosamide exhibits the potential to regulate multiple pathways and to treat other diseases, such as various cancers, bacterial infections, viral infections, and metabolic diseases, in addition to parasite treatment [30,31,32,33,34,35,36,37]. Numerous studies have shown that niclosamide inhibits the growth of various types of tumors [24]. Moreover, niclosamide potentiates the cytotoxic effect of anticancer drugs or radiation therapy [32,37]. Interestingly, niclosamide exhibits minimal effects on normal cells while potentially inhibiting tumor cell proliferation [38]. Cerles O et al. recently showed that niclosamide potentiates the cytotoxicity of oxaliplatin in cancer cells without affecting normal cells [39], suggesting that niclosamide could selectively potentiate anti-tumor effects on cancer cells. More importantly, Cerles O et al. showed that niclosamide ameliorates hypoesthesia and hyperalgesia phenotypes in a mouse model without affecting the antitumor effect of oxaliplatin. These results suggest that niclosamide could protect peripheral neurons without reducing the antitumor effects of chemotherapeutic drugs. Whether niclosamide suppresses neurotoxicity without impairing the antitumor effect of paclitaxel remains to be verified in future work.

Importantly, it has been recently reported that niclosamide rapidly activates PINK1 activity in both HeLa cells and primary cortical neurons [19]. Niclosamide-induced mitochondrial uncoupling, which is required for PINK1 activation, is transient and reversible, and thus niclosamide did not induce mitochondrial damage or cellular toxicity in a previous study [19]. Consistent with these reports, our study shows that niclosamide treatment caused no severe toxicity. First, we found that niclosamide treatment did not interfere with the growth of Drosophila larvae (Figure 1B). Second, we found that dendrite arborization of C4da neurons and thermal nociception were not changed upon niclosamide treatment (Figure 3). Finally, the levels of mitochondrial ROS and mitophagy were not significantly changed by niclosamide in C4da neurons (Figure 4 and Figure 5), suggesting that niclosamide does not induce mitochondrial stress. We confirmed that niclosamide does not increase mitochondrial ROS levels in SH-SY5Y human neuronal cells (Figure 6). Given that PINK1 expression resulted in the alteration of C4da dendrite structure and thermal nociception [16], transient activation of PINK1 by niclosamide may be a particularly effective therapeutic strategy because this compound does not produce severe side effects.

We had previously shown that PINK1 expression alleviates the paclitaxel-induced thermal hypersensitivity phenotype by attenuating mitochondrial dysfunction not by suppressing altered C4da dendrite arborization [16]. In this study, we again observed that paclitaxel-induced alteration of C4da neuron dendrite structure was not affected by niclosamide treatment (Figure 3). In contrast, niclosamide efficiently reduced the paclitaxel-induced increase in mitochondrial ROS levels and the aberrant increase in mitophagy in C4da neurons (Figure 4 and Figure 5). The effect of niclosamide on paclitaxel-induced mitochondrial dysfunction was confirmed in the SH-SY5Y cell line (Figure 6). These results indicate that niclosamide efficiently reduces paclitaxel-induced mitochondrial dysfunction.

Previous studies have revealed that paclitaxel-induced neuronal damage is closely associated with mitochondrial dysfunction. In addition, numerous previous experiments revealed that paclitaxel treatment resulted in structural and functional alterations in mitochondria, such as Ca^2+^ release from mitochondria, an increase in mitochondrial ROS levels, and reduced mitochondrial respiration [6,40,41]. Specifically, Xiao et al. and Zheng et al. revealed that paclitaxel induced mitochondrial dysfunction, including mitochondrial swelling and reduced mitochondrial respiration, in sensory neurons in a rat model [9,12]. Moreover, using a zebrafish model, Cirrincione AM et al. recently showed that an increase in mitochondrial ROS levels is the cause of axon degeneration [10]. These results suggest that mitochondrial dysfunction plays an important role in neuronal damage upon paclitaxel treatment. Thus, a reduction in paclitaxel-induced mitochondrial dysfunction may be an effective strategy for the treatment of peripheral neuropathy. Our results in this study suggest that niclosamide may be a potential candidate for paclitaxel-induced peripheral neuropathy by attenuating mitochondrial dysfunction. Interestingly, niclosamide has been recently shown to ameliorate oxaliplatin-induced tactile hypoesthesia and cold hyperalgesia phenotypes [39]. Given that mitochondrial dysfunction is also closely associated with oxaliplatin-induced peripheral neuropathy, niclosamide may alleviate sensory phenotypes by improving mitochondrial dysfunction. Because mitochondrial dysfunction is considered an important contributor to CIPN [4,6], it would be interesting to determine whether niclosamide has a therapeutic effect on peripheral neuropathy induced by anticancer drugs in addition to paclitaxel.

In summary, our data show that niclosamide ameliorates the thermal hyperalgesia phenotype and mitochondrial dysfunction in paclitaxel-induced peripheral neuropathy in a PINK1-dependent manner. In contrast to PINK1 expression, niclosamide did not induce alterations in sensory neurons or thermal nociception. This study illustrates that niclosamide exhibits therapeutic potential for paclitaxel-induced peripheral neuropathy and that targeting mitochondrial dysfunction may be an effective strategy for the treatment of CIPN.

## Figures and Tables

**Figure 1 biomedicines-10-00863-f001:**
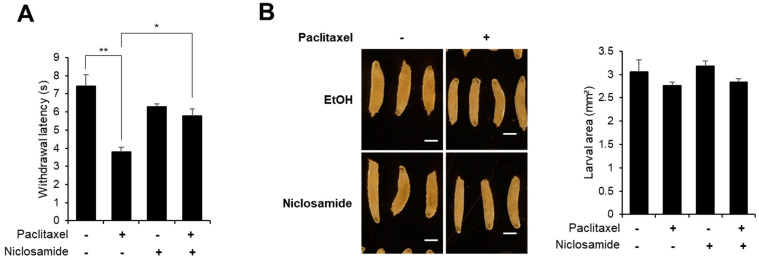
Niclosamide ameliorates the paclitaxel-induced heat hyperalgesia phenotype. (**A**) Thermal nociceptive withdrawal of the *ppk>w^1118^* L3 larvae in response to a 40 °C heat probe after 48 h of exposure to either paclitaxel (20 μM) alone or in combination with niclosamide (250 μM; *n* = 30 per sample). (**B**) Representative images of the larvae of each sample group 120 AEL and treated with either paclitaxel (20 μM) alone or in combination with niclosamide (250 μM) for 48 h (left). The larval area was calculated by the length multiplied by the width of each larva in each sample group, and the results were plotted (*n* ≧ 30 per sample) (right). Scale bars, 1 mm. The experiment was repeated three times, and the results are presented as the mean values with the S.E.M. Significance was determined by one-way ANOVA with Sidák correction. Unless otherwise indicated differences are not significant. ** p* < 0.05; *** p* < 0.01.

**Figure 2 biomedicines-10-00863-f002:**
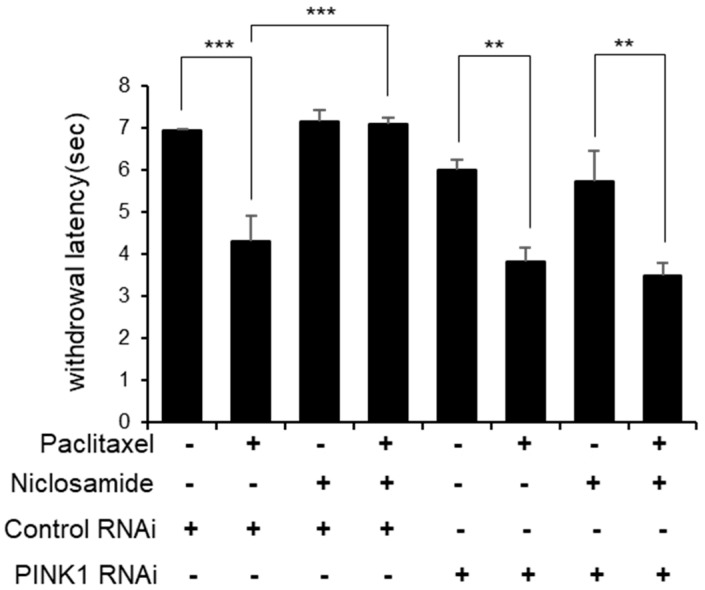
The effect of niclosamide on the paclitaxel-induced heat hyperalgesia phenotype is PINK1-dependent. Thermal nociceptive withdrawal of *ppk>GFP RNAi* and *ppk>PINK1 RNAi* to a heat probe (40 °C) was examined after 48 h of exposure to either paclitaxel (20 μM) alone or in combination with niclosamide (250 μM; *n* = 30 per sample). The experiment was repeated three times, and the results are presented as the mean values with the S.E.M. Significance was determined by one-way ANOVA with Sidák correction. Unless otherwise indicated differences are not significant. ** *p* < 0.01; *** *p* < 0.001.

**Figure 3 biomedicines-10-00863-f003:**
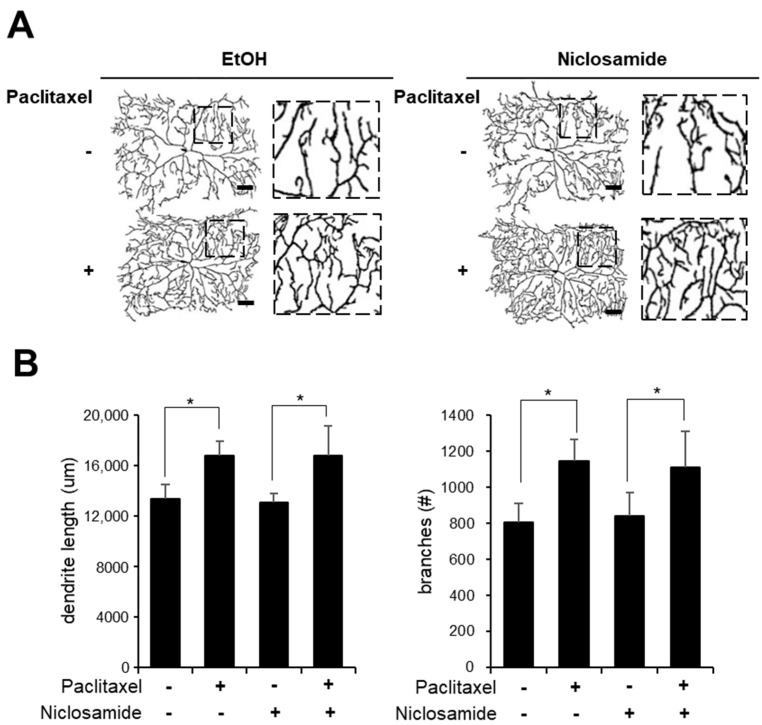
Niclosamide did not reduce paclitaxel-induced arborization in C4da neurons. (**A**) Representative images of C4da neurons at abdominal segment A4 of control L3 larvae (*ppk^1a^>CD4-tdTom*) after 48 h of exposure to paclitaxel (20 μM) either alone or in combination with niclosamide (250 μM). The boxed regions are shown enlarged in the bottom panel. Scale bars, 50 μm. (**B**) Quantification of the length of dendrites and the number of dendritic branch points of C4da neurons; *n* = 3–4 per sample. One C4da neuron per larva was analyzed. The experiment was repeated three times and the results are presented as the mean values with the S.E.M. Significance was determined by one-way ANOVA with Sidák correction. Unless otherwise indicated differences are not significant. * *p* < 0.05.

**Figure 4 biomedicines-10-00863-f004:**
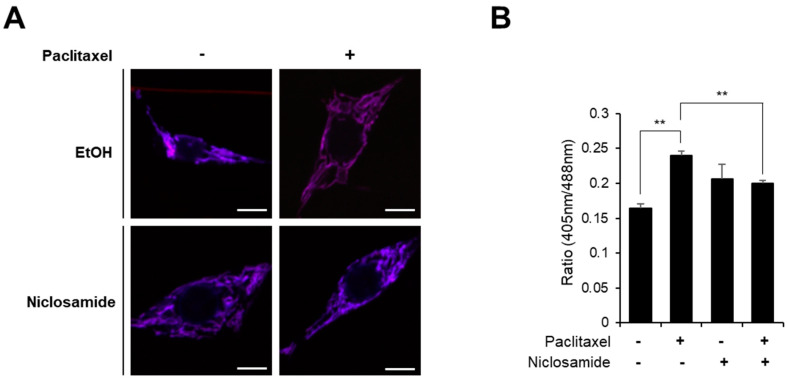
Increases in mitochondrial ROS levels of C4da neurons upon paclitaxel treatment were suppressed by niclosamide. (**A**) Representative fluorescence images of C4da sensory neurons at abdominal segment A4 in L3 larvae expressing the in vivo mitochondrial H_2_O_2_ probe mito-roGFP2-Orp1 (*ppk>mito-roGFP2-Orp1*) with either paclitaxel (20 μM) alone or in combination with niclosamide (250 μM). Scale bars, 5 μm. (**B**) Quantitative analysis in the mitochondrial ROS levels of the C4da sensory neurons in each group (*n* = 5 per group). One C4da neuron per larva was analyzed. The experiment was repeated three times and the results are presented as the mean values with the S.E.M. Significance was determined by one-way ANOVA with Sidák correction. Unless otherwise indicated differences are not significant. ** *p* < 0.01.

**Figure 5 biomedicines-10-00863-f005:**
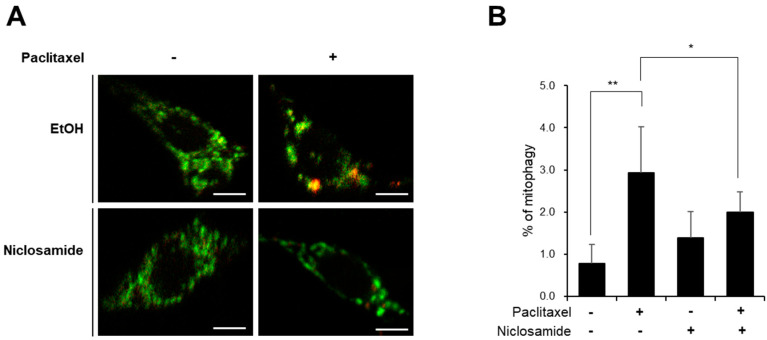
Effect of niclosamide on paclitaxel-induced mitophagy in C4da neurons. (**A**) Representative mt-Keima fluorescence images of C4da sensory neurons at abdominal segment A4 in control L3 larvae (*ppk^1a^>mt-Keima*) treated with paclitaxel (20 μM) either alone or in combination with ni-closamide (250 μM) for 48 h. Scale bars, 5 μm. (**B**) Quantitative analysis of the mitophagy of C4da sensory neurons in each group (*n* = 5 per group). One C4da neuron per larva was analyzed. The results are presented as the mean values, and the error bars represent the SD. Significance was determined by one-way ANOVA with Sidák correction. Unless otherwise indicated differences are not significant. * *p* < 0.05; ** *p* < 0.01.

**Figure 6 biomedicines-10-00863-f006:**
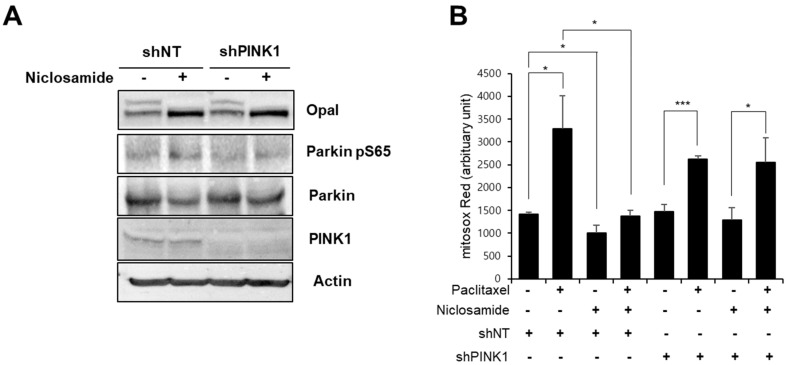
Niclosamide ameliorates paclitaxel-induced mitochondrial dysfunction in SH-SY5Y cells in a PINK1-dependent manner. (**A**) SH-SY5Y cells were treated with niclosamide (10 μM) for 9 h. Cell lysates were prepared and subjected to Western blot analysis using the indicated antibodies. (**B**) SH-SY5Y cells expressing control shRNA (shNT) or PINK1 shRNA (shPINK1) were treated with paclitaxel (1 μM) for 24 h, and then treat with either niclosamide (10 μM) or EtOH for 9 h. After 15 h additional culture, mitochondrial ROS levels were measured by flow cytometry after the cells were stained with MitoSOX Red. Data are presented as the mean ± SD of three biological replicates. Significance was determined by one-way ANOVA with Sidák correction. Unless otherwise indicated differences are not significant. * *p* < 0.05; *** *p* < 0.001.

## Data Availability

Data are all contained within the article.

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
