# Peer review of "The PINK1 Activator Niclosamide Mitigates Mitochondrial Dysfunction and Thermal Hypersensitivity in a Paclitaxel-Induced *Drosophila* Model of Peripheral Neuropathy"

_biomedicines, 2022, doi:10.3390/biomedicines10040863_

Round 1

Reviewer 1 Report

The current study is an intriguing one. Paclitaxel is an anticancer drug. Cancer cells rely on mitochondria to synthesize building blocks during proliferation and bioenergetics, and pharmacological or genetic interference with mitochondrial function is known to attenuate the tumorigenic potential. For example, depletion of cancer cells of mitochondrial DNA impaired their potential to form tumors in vivo, demonstrating the requirement of mitochondrial function for tumorigenesis (Tan et al., 2015). So, mitochondrial dysfunction induced by Paclixatel has a beneficial effect on cancer management.

Moreover, the mitochondrial uncoupler niclosamide and its ethanolamine salt have been shown to possess anti‐tumor efficacy in various preclinical tumor rodent models (Alasadi et al, 2018). However, niclosamide has only minor therapeutic efficacy at its maximal tolerated dose when used as a single drug. Thus, combinatorial treatment with the combination of paclitaxel plus niclosamide could synergize to induce cancer cell death. They could accentuate drug efficacy and thereby provide novel approaches in cancer therapy.

So, I did not understand the reason for the PINK1 reduction by niclosamide of the mitochondrial dysfunction induced by paclitaxel. Has it had a beneficial effect on cancer cells or not?

Maybe there is a different pathway of action for niclosamide in normal cells. But, the various ways had not been discussed by the authors in the study.

Therefore the authors should discuss the mechanism of action from niclosamide in peripheral neuropathy because it is confusing in the manuscript.

Author Response

The current study is an intriguing one. Paclitaxel is an anticancer drug. Cancer cells rely on mitochondria to synthesize building blocks during proliferation and bioenergetics, and pharmacological or genetic interference with mitochondrial function is known to attenuate the tumorigenic potential. For example, depletion of cancer cells of mitochondrial DNA impaired their potential to form tumors in vivo, demonstrating the requirement of mitochondrial function for tumorigenesis (Tan et al., 2015). So, mitochondrial dysfunction induced by Paclixatel has a beneficial effect on cancer management.

Moreover, the mitochondrial uncoupler niclosamide and its ethanolamine salt have been shown to possess anti‐tumor efficacy in various preclinical tumor rodent models (Alasadi et al, 2018). However, niclosamide has only minor therapeutic efficacy at its maximal tolerated dose when used as a single drug. Thus, combinatorial treatment with the combination of paclitaxel plus niclosamide could synergize to induce cancer cell death. They could accentuate drug efficacy and thereby provide novel approaches in cancer therapy.

So, I did not understand the reason for the PINK1 reduction by niclosamide of the mitochondrial dysfunction induced by paclitaxel. Has it had a beneficial effect on cancer cells or not?

Maybe there is a different pathway of action for niclosamide in normal cells. But, the various ways had not been discussed by the authors in the study.

Therefore the authors should discuss the mechanism of action from niclosamide in peripheral neuropathy because it is confusing in the manuscript.

Response: We understand the reviewer’s concern about the possible influence of niclosamide on cancer treatment by paclitaxel. We agree with the reviewer that paclitaxel-induced mitochondrial dysfunction of cancer cells could be beneficial for cancer treatment. However, given that mitochondrial dysfunction of sensory neurons is an important cause of peripheral neuropathy, it is necessary to develop a strategy that can either selectively prevent mitochondrial dysfunction of sensory neurons or improve mitochondrial function after chemotherapy completion.

As the reviewer pointed out, numerous studies have shown that niclosamide inhibits the growth of various types of tumors (Reviewed by Chen W et al, 2017, Cell Signal, doi: 10.1016/j.cellsig.2017.04.001). Moreover, niclosamide has been shown to potentiate the cytotoxic effects of anticancer drugs or radiation therapy (You S et al, 2014, Mol Cancer Ther, DOI: 10.1158/1535-7163.MCT-13-0608; Lee SL et al, 2014, Biomed Pharmacother, DOI: 10.1016/j.biopha.2014.03.018). Interestingly, niclosamide exhibits minimal effect on normal cells while potentially inhibiting tumor cell proliferation (Jin Y et al, 2010, Cancer Res, DOI: 10.1158/0008-5472.CAN-09-3950). Cerles O et al recently showed that niclosamide potentiates the cytotoxicity of oxaliplatin in cancer cells without affecting normal cells (Cerles O et al, 2017, Mol Cancer Ther, doi: 10.1158/1535-7163.MCT-16-0326), suggesting that niclosamide could selectively potentiate antitumor effects on cancer cells. More importantly, Cerles O et al showed that niclosamide ameliorates hypoesthesia and hyperalgesia phenotypes in a mouse model without changing the antitumor effect of oxaliplatin. These results suggest that niclosamide could protect peripheral neurons without reducing the antitumor effects of chemotherapeutic drugs. However, whether niclosamide suppress neurotoxicity without impairing the antitumor effects of paclitaxel remains to be verified in future work.

We have revised the Discussion section to explain these points.

Reviewer 2 Report

Paclitaxel is an anticancer drug that induces a dose-limiting peripheral neuropathy. Using a previously established paclitaxel-induced thermal hypersensitivity model in Drosophila larvae, the authors demonstrated that niclosamide significantly ameliorated the paclitaxel-induced thermal hypersensitivity phenotype via PINK1. Therefore, the authors concluded that niclosamide has therapeutic potential in paclitaxel-induced peripheral neuropathy. This is good work based largely on the findings of a previously published article by the same group. Background, methods and results are well presented and discussed. However, one concern is that, although the mean test shows that the probability of the null hypothesis is low, the authors used only a small number of larvae (two or three per group) for the evaluation of the mean differences, and perhaps the precision of the observed differences are not robust enough. Therefore, it would be advisable that the authors justify the reason for that number.

Author Response

Paclitaxel is an anticancer drug that induces a dose-limiting peripheral neuropathy. Using a previously established paclitaxel-induced thermal hypersensitivity model in Drosophila larvae, the authors demonstrated that niclosamide significantly ameliorated the paclitaxel-induced thermal hypersensitivity phenotype via PINK1. Therefore, the authors concluded that niclosamide has therapeutic potential in paclitaxel-induced peripheral neuropathy. This is good work based largely on the findings of a previously published article by the same group. Background, methods and results are well presented and discussed. However, one concern is that, although the mean test shows that the probability of the null hypothesis is low, the authors used only a small number of larvae (two or three per group) for the evaluation of the mean differences, and perhaps the precision of the observed differences are not robust enough. Therefore, it would be advisable that the authors justify the reason for that number.

Response: We apologize for the confusion caused by the incorrect description of the experimental procedure. To acquire statistically meaningful data, we repeated the experiment several times and analyzed a sufficient number of larvae.

For the C4da neuron dendrite analysis in Figure 3, three or four larvae per group were analyzed. We repeated the dendrite analysis experiment three times, and the results are presented as the mean value with the standard error of the mean (S.E.M.).

For the mitochondrial ROS analysis of C4da neurons in Figure 4, four or five larvae per group were analyzed. We also repeated the mitochondrial ROS analysis three times, and the results are presented as the mean value with the S.E.M.

For the measurements of mitophagy in C4da neurons in Figure 6, we analyzed five larvae per group, and the results are presented as the mean value ± SD.

For the thermal nociception assay in Figure 1 and 2, thirty larvae per group were analyzed. We also repeated the thermal nociception assay three times, and the results are presented as the mean value with the S.E.M.

We attached the original data set as “Original data set” for the reviewers (Please see the attached file). In addition, we revised the Material and Methods section and figure legends to clearly explain how the results were obtained.

Reviewer 3 Report

The authors aim to show the therapeutic potential of small-molecule PINK1 activator niclosamide for paclitaxel-induced peripheral neuropathy using the Drosophila larvae model and human neuroblastoma SH-SY5Y cell line. While interesting, some points require to clarify:

  1. The background for investigating C4da neurons should be provided.
  2. The samples size needs to be clarified. For example: in Figure 3B “n = 3-4 per sample”; in Figure 4B “n = 5 per group”; in Figure 5B “n = 5 per group”. The number of neurons counted in the number of specific power fields, in the number of larvae should be clarified. In Figure 6B, biological replicate and technical replicate should be shown. Original mitochondria ROS level should be shown instead of fold.
  3. The description of statistical analysis needs to be clarified. "To compare three or more groups, we performed one-way ANOVA with Sidák correction."However, through the current manuscript, all statistical analyses were shown pair-wised.
  4. “Niclosamide suppressed paclitaxel-induced mitochondrial dysfunction” in human SH-SY5Y cells in a PINK1-dependent manner.” is overstated. The current results showed only the mitochondrial ROS level. Noted that PINK1-knockdown also reduces paclitaxel-induced mitochondrial ROS levels. In addition, a dose-dependent response will be appreciated.

Author Response

The authors aim to show the therapeutic potential of small-molecule PINK1 activator niclosamide for paclitaxel-induced peripheral neuropathy using the Drosophila larvae model and human neuroblastoma SH-SY5Y cell line. While interesting, some points require to clarify:

  1. The background for investigating C4da neurons should be provided.

Response: In Drosophila larvae, class IV dendritic arborization (C4da) neurons are sensory nociceptors that are responsible for behavioral responses to various noxious stimuli, including high temperatures [17,18]. We revised the Introduction and Results section to clearly describe the reason for investigating C4da neurons as shown below.

(Line 58) Interestingly, PINK1 expression did not suppress paclitaxel-induced alterations to the sensory dendrite structure of class IV dendritic arborization (C4da) neurons, which are responsible for behavioral responses to various noxious stimuli, including high temperatures [17,18]. However, paclitaxel-induced increases in both mitochondrial ROS and mitophagy were significantly suppressed by PINK1 expression [16], suggesting that PINK1 can reduce the mitochondrial dysfunction induced by paclitaxel.

(Line 209) To verify that niclosamide suppressed the paclitaxel-induced thermal hyperalgesia phenotype through PINK1, we examined the effect of niclosamide on the specific knockdown of PINK1 expression in C4da neurons because C4da neurons are responsible for behavioral responses to high temperatures [17,18].

(Line 238) Alterations in the sensory dendritic structure of C4da neurons are associated with paclitaxel-induced peripheral neuropathy [16,21].

  1. The samples size needs to be clarified. For example: in Figure 3B “n = 3-4 per sample”; in Figure 4B “n = 5 per group”; in Figure 5B “n = 5 per group”. The number of neurons counted in the number of specific power fields, in the number of larvae should be clarified. In Figure 6B, biological replicate and technical replicate should be shown. Original mitochondria ROS level should be shown instead of fold.

Response: We apologize for insufficient description of the experimental procedure. In Figures 3, 4, and 5, we analyzed one C4da neuron per larva. In addition, in Figures 3 and 4, we repeated the experiment three times, and the results are presented as the mean with the standard error of the mean (S.E.M.). We carefully revised the Materials and Methods section and figure legends to clearly describe these points as shown below.

In addition, according to the reviewer’s suggestion, Figure 6B now shows the mitoSox Red fluorescence intensity instead of the fold difference.

(Line 111) The dorsal projection of one C4da neuron per larva from abdominal segment A4 was analyzed. At least 3-4 larvae in each genotype group were analyzed. The dendrite image analysis was repeated three times, and the results are presented as the mean value with the standard error of the mean (S.E.M.).

(Line 120) One C4da neuron per larva from abdominal segment A4 was analyzed. The 405-nm/488-nm fluorescence intensity was obtained using Zeiss Zen software, and five larvae in each group were measured. The experiment was repeated three times, and the results are presented as the mean value with the S.E.M.

(Line 131) One C4da neuron per larva from abdominal segment A4 was analyzed.

(Line 161) Data are presented as the mean ± SD or mean ± S.E.M. as indicated in the figure legend.

(Line 262) One C4da neuron per larva was analyzed. The experiment was repeated three times, and the results are presented as the mean value with the S.E.M.

(Line 275) One C4da neuron per larva was analyzed. The experiment was repeated three times, and the results are presented as the mean value with the S.E.M.

(Line 305) One C4da neuron per larva was analyzed.

  1. The description of statistical analysis needs to be clarified. "To compare three or more groups, we performed one-way ANOVA with Sidák correction."However, through the current manuscript, all statistical analyses were shown pair-wised.

Response: We apologize for the incorrect description of the statistical analysis. We used one-way ANOVA in this study to compare differences between groups. To avoid possible confusion about the statistical analysis methods, we have indicated only significant p values and removed all indications of NS. Accordingly, we state that “Unless otherwise indicated, differences are not significant”.

In addition, we have revised all of the graphs to avoid possible confusion.

We revised the Materials and Methods section and figure legends to clarify the statistical analysis as shown below.

(Line 160) 2.9. Statistical analysis

The results are presented as the mean ± SD or mean ± S.E.M. as indicated in the figure legend. One-way ANOVA with Sidák correction was used to compare three or more groups. All data were statistically analyzed using GraphPad Prism software (version 8.0: GraphPad Software, Inc., La Jolla, CA, USA). A P value of < 0.05 was considered statistically significant.

(Line 197) Significance was determined by one-way ANOVA with Sidák correction. Unless otherwise indicated, differences are not significant. *p < 0.05; **p < 0.01.

(Line 233) Significance was determined by one-way ANOVA with Sidák correction. Unless otherwise indicated differences are not significant. **p < 0.01; *** p < 0.001; **** p < 0.0001.

(Line 263) Significance was determined by one-way ANOVA with Sidák correction. Unless otherwise indicated, differences are not significant. *p < 0.05.

(Line 276) Significance was determined by one-way ANOVA with Sidák correction. Unless otherwise indicated, differences are not significant. ** p < 0.01.

(Line 306) Significance was determined by one-way ANOVA with Sidák correction. Unless otherwise indicated, differences are not significant. *p < 0.05; **p < 0.01.

(Line 337) Significance was determined by one-way ANOVA with Sidák correction. Unless otherwise indicated, differences are not significant. * p < 0.05.

  1. “Niclosamide suppressed paclitaxel-induced mitochondrial dysfunction” in human SH-SY5Y cells in a PINK1-dependent manner.” is overstated. The current results showed only the mitochondrial ROS level. Noted that PINK1-knockdown also reduces paclitaxel-induced mitochondrial ROS levels. In addition, a dose-dependent response will be appreciated.

Response: According to the reviewer’s suggestion, we have revised the corresponding sentence as shown below.

(Line 325) These results suggest that niclosamide efficiently suppressed the paclitaxel-induced increase in mitochondrial ROS in a PINK1-dependent manner in human cells.